# Self-Supervised Masked Autoencoders for Prostate Cancer Segmentation via Learned Representations

## Abstract

Prostate cancer incidence is rising at an estimated rate of 6.7% annually. MRI guided biopsy of suspected lesions can improve diagnosis but requires precise delineation of the prostate gland and lesions. Manually delineating lesions is labor intensive and remains challenging due to their morphological similarity to surrounding healthy tissue. In this study, we propose a self-supervised learning framework to learn rich anatomical and pathological representations from bi-parametric MRI (bpMRI) scans. Specifically, we leverage a Masked Autoencoder (MAE) architecture trained on multimodal bpMRI to capture context-aware features of the prostate region through a novel lesion-aware masking strategy. The pretrained encoder is then fine-tuned for lesion segmentation using a lightweight decoder augmented with skip connections from the MAE. Our fine-tuning strategy incorporates a balanced dataset and a hybrid loss function to address class imbalance.The proposed approach outperforms state-of-the-art segmentation methods, achieving over a 10% improvement in Dice score on the held-out test set.

## 1 Introduction

Prostate cancer (PCa) is the second most commonly diagnosed cancer in men globally, with over 1.2 million new cases annually and more than 350,000 deaths each year. In the United States alone, the American Cancer Society estimates approximately 313,780 new cases and 35,770 deaths in 2025, accounting for roughly 15.4% [1], of all new cancer diagnoses in men .

Bi-parametric MRI (bpMRI), which includes T2-weighted (T2W) and diffusion-weighted imaging (DWI), has become an essential tool in the prostate cancer (PCa) diagnostic pathway. bpMRI improves lesion detection within the prostate gland, enables targeted biopsies, and is widely regarded as the preferred non-invasive approach for PCa risk stratification.

Accurate segmentation of the prostate and suspicious lesions can guide targeted biopsies and assist in automated lesion grading, thereby improving diagnostic precision. At present, these tasks rely heavily on the expertise and experience of clinicians, contributing to high workload and leading to variability in outcomes.

Lesion segmentation in prostate MRI is an inherently challenging task due to combination of clinical, anatomical, and technical factors. Accurate detection of both clinically significant and non-significant lesions is essential to improve diagnostic performance, yet several barriers persist. Subtle intensity differences between tumors and surrounding healthy tissue make lesion boundaries difficult to delineate, while anatomical complexity further complicates segmentation, as tumors often exhibit irregular shapes, ill-defined borders, and multifocal patterns dispersed across the gland. In addition, inter-reader variability, severe class imbalance, and variability across scanners and acquisition protocols amplify these challenges. Collectively, these factors highlight the need for

robust, automated methods capable of learning discriminative representations from MRI data to achieve reliable lesion segmentation.

In view of this, the following study introduces a self-supervised (SSL) approach towards learning pathology specific representations, we introduce a Masked Autoencoder (MAE) [2], architecture on bpMRIs of the prostate. The multimodal approach allows the model to learns joint latent representations that capture both structural anatomy and pathological patterns.

Below, we highlight the key contributions and generalizable insights of our approach:

- **Lesion-aware masking in self-supervised pretraining encourages clinical relevance in learned representations**: By prioritizing the retention of lesion-containing regions during masked autoencoding, the model learns to encode semantically meaningful features around pathologies—without explicit labels. This approach can generalize to other sparse abnormalities.
- **An effective way to learn from limited labelled data**: Our approach introduces an encoder-heavy architecture that leverages unlabeled data through self-supervised learning to capture rich, domain-specific representations. To adapt these representations to downstream tasks, we pair the encoder with a lightweight decoder tailored for efficient fine-tuning. This design enables rapid generalization to multiple medical imaging tasks, particularly in settings with scarce annotated data.
- **Balanced data construction addresses dual imbalance in medical tasks**: Our approach introduces a tailored data sampling strategy that enables the encoder to learn from the full, naturally distributed dataset during pretraining, while the fine-tuning stage employs a balanced subset to address the dual challenge of underrepresented positive samples (lesion-containing slices) and spatial sparsity of lesions within each image. This design ensures robust representation learning and improved segmentation performance on clinically relevant regions.

## 2 Methods

This section outlines our two-phase training pipeline (Figure 1), Phase 1 involves pretraining a custom MAE on 4,636 volumetric bpMRI exams, the details for the dataset can be found in Appendix A.1. The MAE is trained to reconstruct masked portions of the input using a **lesion-aware masking strategy**, which prioritizes the retention of lesion-containing patches to enforce richer feature learning in clinically relevant regions. The encoder learns high-level representations of the prostate anatomy and potential lesion patterns without requiring ground-truth annotations, (see Appendix A.2).

In Phase 2, we fine-tune the pretrained encoder for supervised segmentation. A lightweight decoder head is appended to the encoder to form a complete prediction network. The segmentation model is optimized using a hybrid loss function that combines focal loss and Dice loss to handle class imbalance and optimize overlap-based accuracy. The overall architecture is tailored to the spatial resolution and anatomical structure of the prostate region (see Appendix A.3).

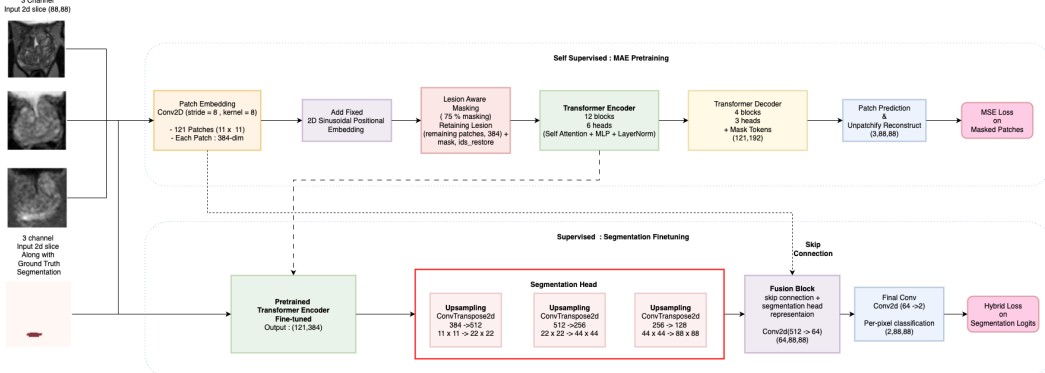

Figure 1: Multimodal Two-phase training pipeline

## 3   Results

To quantitatively evaluate the performance of our proposed lesion segmentation framework we reserve a dedicated test set comprising 693 prostate bpMRI volumes, yielding a total of 44,352 2D slices. These volumes are strictly held out during both self-supervised pretraining and supervised fine-tuning to ensure an unbiased assessment of generalization capability, segmentation visualisation can be seen in Appendix A.4. Table 1 reports the Dice Similarity Coefficient (Dice) and the Intersection over Union (IoU), computed on a per-slice basis compared against state of art methods.

Table 1: Comparison to state-of-art approachs

| Method | Dice Score | IoU |
|---|---|---|
| UNet [3] | 0.2031 | 0.1429 |
| VNet [4] | 0.1891 | 0.1264 |
| Attention-UNet [5] | 0.2203 | 0.1937 |
| TransUnet [6] | 0.2601 | 0.2122 |
| SwinUnet [7] | 0.2891 | 0.2311 |
| Our Approach | **0.4213** | **0.3785** |

## 4   Discussion and Future Work

A persistent bottleneck in medical image analysis remains the lack of large, annotated datasets. Even with increasingly powerful model architectures, the scarcity of expert-annotated labels limits the performance and generalizability of supervised approaches. A common workaround is to adopt transfer learning from models pretrained on generic datasets like ImageNet. However, such models often fail to generalize well to medical domains due to a significant domain shift. Features learned from natural images do not adequately capture the complex texture, anatomy, and modality-specific patterns in medical scans.

Recognizing these gaps, our approach leverages self-supervised learning (SSL) to pretrain on un-labelled prostate bpMRI data. This allows the model to learn prostate-specific anatomical and pathological features directly from the domain of interest, without reliance on external data distributions.

Lesion segmentation in the prostate often resembles a concealed object segmentation task. Lesions tend to be amorphous, poorly contrasted, and lack consistent structure, making them difficult to learn through traditional pattern-based methods under supervised settings. Conventional CNN-based models, which rely heavily on localized features and spatial hierarchies, struggle to generalize across the morphological variability of prostate lesions. In contrast, we employ a transformer-based encoder, specifically a Masked Autoencoder (MAE) which captures global context and non-local dependencies, making it more suitable for learning nuanced lesion features across diverse patient populations. By integrating skip connections from the MAE encoder into a lightweight CNN-based decoder, our finetuned segmentation head combines rich contextual embeddings with localized upsampling, enabling precise and robust lesion delineation.

Our proposed framework shows strong performance in segmenting both clinically significant and non-significant prostate lesions, highlighting its potential to enhance diagnostic precision across the spectrum of prostate cancer. Moving forward, we aim to extend this work toward anatomy-specific generalizability by developing task-specific decoder heads for downstream applications, such as identifying clinically significant lesions and delineating finer prostate subregions. These extensions will further advance the clinical relevance of our approach and support its scalable deployment in real-world settings

## 5   Potential Negative Societal Impact

While our proposed framework demonstrates promising performance in prostate lesion segmentation, several potential negative societal impacts warrant careful consideration. Automated segmentation errors could lead to missed cancerous lesions or false positives, potentially delaying treatment or

causing unnecessary biopsies. False negatives are particularly dangerous as they may provide false reassurance to patients with actual cancer. Furthermore, clinicians may overrely on the system, reducing critical human oversight in diagnosis, a concern amplified by the "black box" nature of deep learning models, which makes it difficult for clinicians to understand why certain predictions were made and, in cases of misdiagnosis, creates challenges in determining accountability among model developers, clinicians, and institutions. The model may perform poorly on underrepresented patient populations if the training data lacks diversity across different ethnicities, age groups, and scanner types, and such performance disparities could exacerbate existing healthcare inequities, with certain demographic groups receiving inferior diagnostic accuracy. Additionally, patients may not be adequately informed about AI involvement in their diagnosis, raising concerns about informed consent and transparency. To mitigate these risks, we emphasize that this system is designed to assist, not replace, clinical expertise. We strongly recommend rigorous external validation across diverse patient populations and clinical settings, continuous monitoring of model performance post-deployment, and clear communication with patients regarding AI-assisted diagnosis. Future work should prioritize model interpretability and establish robust clinical validation protocols before real-world deployment.

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

# A  Appendix

## A.1  Dataset & Preprocessing

The dataset used in this study comprised 4,636 3T bpMRI volumetric scans, collected with institutional review board approval and patient consent. Each scan includes axial T2W imaging, DWI, and apparent diffusion coefficient (ADC) maps. High b-value images (b = 1500 s/mm²) were calculated from acquired b = 50 and b = 1000 s/mm² data following standard protocols.

All scans were co-registered to the T2W space to ensure spatial alignment across modalities. The volumes were then resampled to a standardized size of (192 × 192 × 64) voxels and intensity-normalized.

Given our focus on lesion identification within the prostate region, we trained a 3D U-Net architecture from scratch to automatically segment the prostate gland. This segmentation model achieved a Dice coefficient of 0.90 for prostate region identification, demonstrating robust performance for region-of-interest (ROI) extraction. Using the automated prostate segmentation masks, we cropped axial regions around the prostate with 8-pixel padding along both height and width dimensions to ensure complete prostate coverage while accounting for potential segmentation uncertainties. Following ROI extraction, the volumetric scans were resized to (88 × 88 × 64) voxels.

For training, the 3D volumes were processed as 2D slices of size (88 × 88) along the z-direction. Each slice comprised three co-registered modalities—T2W, ADC, and DWI—which were concatenated along the channel dimension, resulting in a 3-channel input tensor of dimensions (3 × 88 × 88). This 2D slice-based approach enabled efficient processing while preserving the multiparametric information essential for lesion characterization.

## A.2 Representation Learning in multimodal setting

The MAE framework consists of two core components: a Vision Transformer (ViT)-based [8], encoder and a lightweight Transformer decoder, optimized to learn anatomical and pathological representations from 2D axial slices extracted from 3D volumetric scans. Each input slice is represented as a 3-channel image, these three modalities are stacked channel-wise to form a tensor of shape (3 × 88 × 88). During training, all three channels are treated equally and passed through a shared convolutional patch embedding layer, which projects the full input into patch tokens without distinguishing between modalities. However, because the model is trained to reconstruct all three channels jointly, it implicitly learns inter-modality correlations.

This multi-modal joint reconstruction enables the model to:

- Leverage structural consistency in T2W images
- Learn contrast sensitivity from ADC values
- Attend to high-signal specific regions from the DWI

This cross-modal representation learning serves as critical step for the downstream lesion segmentation, to learn the subtle differences between these modalities.

The input image is split into 8×8 non-overlapping patches using a convolutional embedding layer. For an 88×88 slice, this results in 11×11=121 patches. Each patch is projected into a high-dimensional embedding (384 dimensions) and enriched with a 2D sinusoidal positional encoding that preserves spatial structure.

The encoder comprises 12 Transformer blocks, each with multi-head self-attention and feedforward MLPs. It operates on only the visible (unmasked) patches, enabling the model to learn from context. A learnable [MASK] token is later used by the decoder to predict masked patches.

In conventional MAE training, a fixed proportion of input patches are randomly masked, and the model is trained to reconstruct them using only the visible patches. However, this randomness can result in lesion-containing patches being masked out entirely — which is suboptimal for medical images, where pathologies may occupy a very small portion of the image. To improve focus on clinically meaningful regions, we introduce **lesion-aware masking** designed specifically to enhance the MAE's ability to learn informative representations of prostate lesions from multi-modal MRI.

Our lesions aware masking uses binary lesion annotations to bias the patch sampling process, without introducing label supervision into the loss. Each input slice is accompanied by a binary lesion mask $M \in {0, 1}^{H \times W}$ where 1 denotes the lesion pixel. We downsample $M$ to patch resolution using average pooling with a kernel and stride equal to the patch size, (8x8) for our case. This yields a patch-level lesion score matrix $M' \in [0, 1]^{H_p \times W_p}$ where each entry represents the fraction of lesion pixels in a patch. We convert $M'$ into sampling weights using the following transformation (1)

$$w_{ij} = 1 + \alpha . M'_{ij} \tag{1}$$

Here $\alpha$ is a hyperparameter that controls the emphasis on lesion patches, we set it to $\alpha$ to 10 for our implementation. This ensures that lesion-containing patches are preferentially retained during MAE training, while the surrounding context is masked and must be reconstructed. As a result:

- The model is forced to reconstruct anatomical context around the lesion

- It still learns from both healthy and pathological regions

- No lesion labels are used in the reconstruction loss — only in guiding the mask.

Following the lesion aware masking, the decoder reconstructs the complete slice (all three channels) by predicting the pixel values of masked patches. It consists of 4 lightweight Transformer blocks followed by a linear projection layer that maps each token back to a flattened 8×8×3 patch. The reconstruction loss is computed only on masked patches using mean squared error (MSE) (2). Before computing the reconstruction loss, each target patch is independently normalized to zero mean and unit variance. This patch-wise normalization ensures consistent gradient scaling across modalities and patch intensities, mitigating training instability due to high dynamic range in multi-modal inputs, like the bright DWI lesions and low-signal ADC regions. It also encourages the model to learn relative structural patterns rather than raw intensities, which improves generalizability across varying scan conditions.

$$L_{MAE} = \frac{1}{\sum m_i} \cdot \sum_i^N m_i \cdot ||\hat{x}_i - x_i||^2 \tag{2}$$

where $m_i \in 0, 1$ is the binary mask for the patch i , $\hat{x}$ and $x$ are the reconstructed and original patches respectively.

We train the MAE for 250 epochs using the AdamW optimizer with a cosine learning rate scheduler, Figure 2 shows the MAE loss curve. The training data for this phase includes all axial slices from all training volumes without filtering or balancing. This ensures that the MAE is exposed to the natural distribution of normal and pathological anatomy, including slices with no lesions.

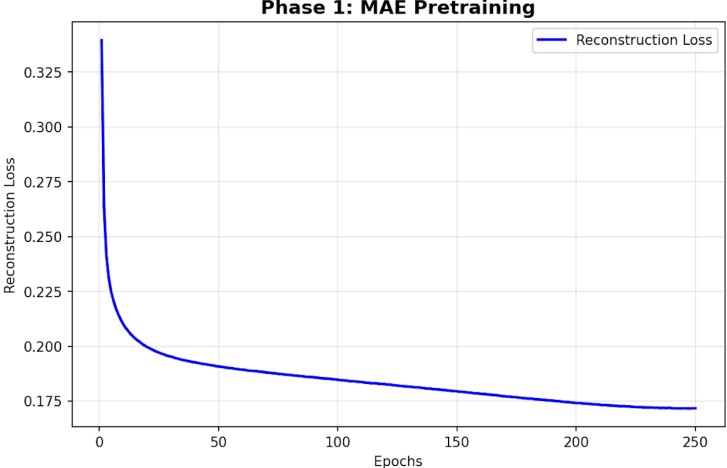

Figure 2: MAE training loss curve

This phase results in a pretrained encoder with structural priors to build lesion specific understanding allowing the model to learn clinical relevance in an unsupervised setting, Figure 3 & 4.

## A.3  Finetuning for Lesion Segmentation

Following the SSL pretraining, we fine-tune the encoder for the prostate lesion segmentation task using the annotated slice. The pretrained encoder is frozen initially and then fine-tuned along with a

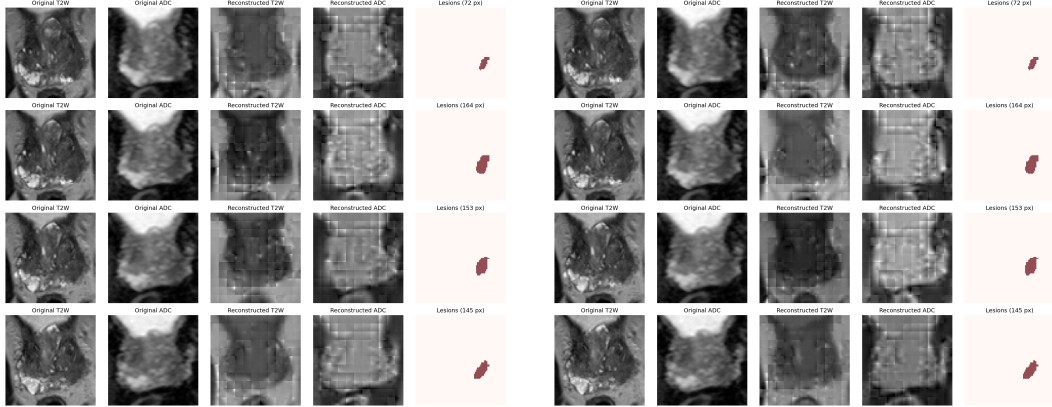

Figure 3: MAE reconstruction 100 epoch         Figure 4: MAE reconstruction 250 epoch

lightweight decoder. This phase leverages the rich representations learned during Phase 1 to improve lesion detection performance.

To recover full-resolution predictions from the patch-wise latent representations, the decoder is composed of 3 upsampling CNN blocks implemented to expand the spatial resolution from patch-level (11×11) back to full image size (88×88). We introduce a fusion block that concatenates upsampled decoder features with skip features from the encoder's patch embedding layer, to inject high-level spatial features from the encoder layer. The decoder and skip features are fused via a 3×3 convolution and passed through a final 1×1 convolutional layer to produce the two-class output mask.

We use differential learning rates while finetuning the encoder and decoder head, a lower learning rate, of $5.00e - 06$, for the pretrained encoder to preserve learned features while a higher of learning rate, of $5.00e - 05$, for the newly initialized decoder to adapt to the segmentation task.

Along with sharing close morhpoholigical features between benign and malignant structures, PCa lesion an extended challenge of the class imbalance problem at two levels, most bpMRI slices contain no lesions and even with slices containing the lesions, lesions occupy an extremely small fraction of the pixels.

To address the class imbalance issue, we construct a balanced lesion sampling strategy along with a hybrid loss function. While fintuning, include all slices with lesions, provided they contain at least a minimum number of lesion pixels, set to be 5 pixels for our study. We randomly sample non-lesion slices to make up the remainder of the training set, maintaining a 50%–50% ratio of lesion vs. non-lesion slices. This ensures that the model is exposed to a meaningful number of lesion samples while still learning to differentiate lesions from healthy tissue. To further combat pixel-level class imbalance, we employ a hybrid loss function (3) that combines focal loss (4) and dice loss (5)

$$L_{semgnetation} = \lambda_{focal} \cdot L_{focal} + \lambda_{dice} \cdot L_{dice}, \quad \text{where } \lambda_{focal} = \lambda_{dice} = 0.5 \tag{3}$$

here, $L_{focal}$ is calculated as follows:

Let the cross-entropy loss be defined as:

$$\text{CE}(x, y) = -\log p_t, \quad \text{where } p_t = \begin{cases} p & \text{if } y = 1 \\ 1 - p & \text{if } y = 0 \end{cases}$$

Then, the focal loss is:

$$L_{\text{focal}} = \frac{1}{N} \sum_{i=1}^{N} \alpha \cdot (1 - p_t^{(i)})^{\gamma} \cdot \text{CE}(x^{(i)}, y^{(i)}) \tag{4}$$

where $N = B \cdot H \cdot W$ is the number of pixels in the batch, $\alpha = 0.25$, $\gamma = 2.0$ (selected hyperparameters) and $p_t^{(i)}$ is the model's predicted probability

265    $L_{dice}$ for class $c \in \{0, 1\}$ (background and lesion) is computed as:

$$\mathcal{L}_{\text{dice}} = 2 - \sum_{c=0}^{1} \frac{2 \cdot \sum\limits_{i} p_c^{(i)} \cdot y_c^{(i)} + \epsilon}{\sum\limits_{i} p_c^{(i)} + \sum\limits_{i} y_c^{(i)} + \epsilon} \tag{5}$$

266    where $p_c^{(i)}$ is the predicted probability for class $c$ at pixel $i$, $y_c^{(i)}$ is the one-hot encoded ground truth
267    for class $c$ at pixel $i$ and $\epsilon$ is a smoothing constant, set to $10^{-5}$ to avoid division by zero

268    The AdamW optimizer with a cosine annealing learning rate schedule is used to stabilize training.
269    Model performance is monitored using the dice score on a held-out validation set, and early stopping
270    is applied if no improvement is observed over 15 consecutive epochs. Following this two-phase
271    training strategy - combining self-supervised pretraining with supervised fine-tuning - our approach
272    achieves strong performance and outperforms several recent state-of-the-art methods for prostate
273    lesion segmentation.

## A.4    Results for Segmentation

275    We report standard overlap-based segmentation metrics, including the Dice Similarity Coefficient
276    (Dice) (6) and the Intersection over Union (IoU) (7), computed on a per-slice basis.

$$\text{Dice} = \frac{2 \times \text{Area of Overlap}}{\text{Total Area}} = \frac{2 \cdot |A \cap B|}{|A| + |B|} \tag{6}$$

$$\text{IoU} = \frac{\text{Area of Overlap}}{\text{Area of Union}} = \frac{|A \cap B|}{|A \cup B|} \tag{7}$$

277    where A is the predicted segmentation mask and B is the ground truth mask

278    Figure 5 and Figure 6, show the qualitative results on the test set samples showing T2W, ADC, and
279    DWI inputs with ground truth (GT) and model predictions for lesion segmentation.

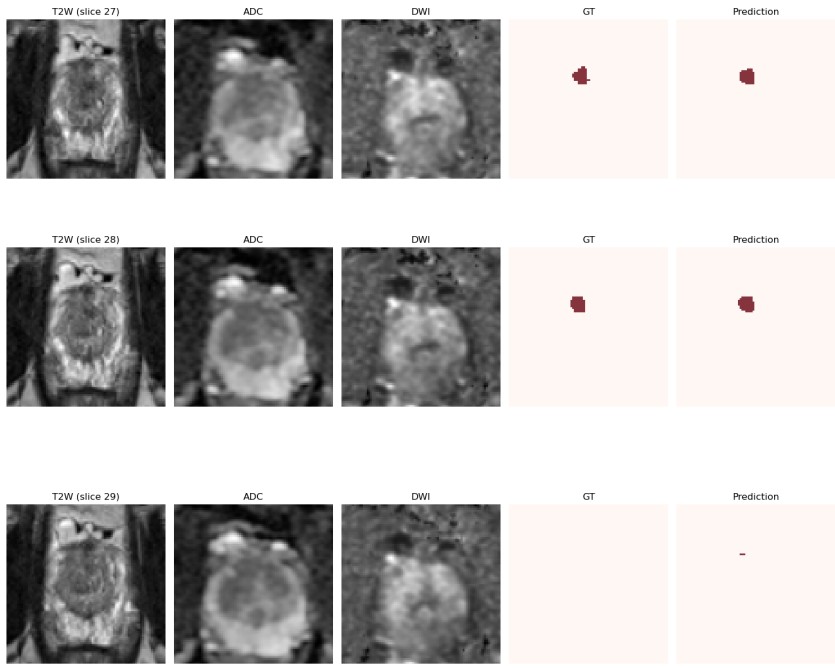

Figure 5: Qualitative Segmentation results

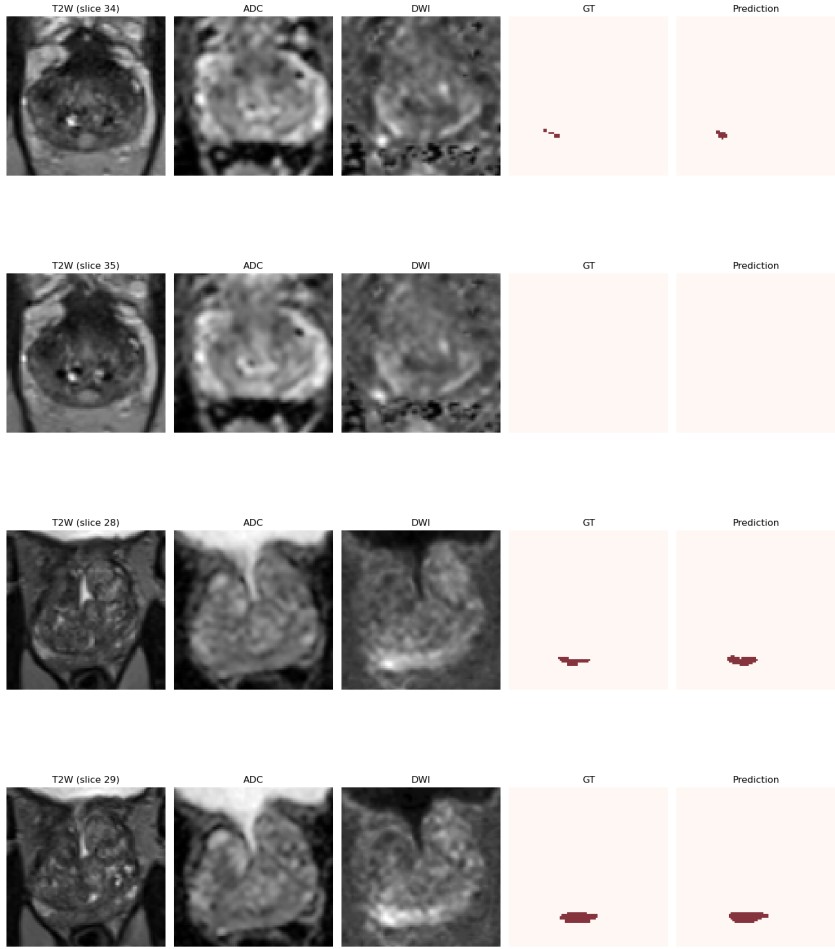

Figure 6: Qualitative Segmentation results (Part 2)

