# OpenReview forum: "Self-Supervised Masked Autoencoders for Prostate Cancer Segmentation via Learned Representations"
_EurIPS.cc/2025/Workshop/MedEurIPS — EurIPS 2025 Workshop MedEurIPS Submission_

### Official Review · Reviewer_GPoL · 2025-10-21
**A clinically relevant approach constrained by questionable methodology and limited novelty.**

**Rating:** 4
**Confidence:** 4

**Review:**

Summary:

This paper presents a two-phase framework for prostate cancer lesion segmentation from bpMRI. First, a Masked Autoencoder (MAE) is pre-trained using a lesion-aware masking strategy to learn domain-specific features. Then, the encoder is fine-tuned with a lightweight decoder for supervised segmentation.

Advantages:

1. Tackles an important clinical problem with clear motivation and presentation.

2. The proposed method is simple and well-explained.

Disadvantages:

Major:

1. The paper claims to propose a self-supervised framework suited for data-scarce clinical settings. However, it is unclear how the referenced 3D UNet used for producing labels on the dataset is trained. Is the reported 0.90 Dice score measured against ground truth annotations? If so, the method indirectly relies on labelled data, which contradicts the self-supervised premise. Rather, it aligns more with a weakly- or semi-supervised approach.

2. The entire workflow depends on an initial prostate gland segmentation performed by a 3D U-Net to define the region of interest. The authors report a Dice of 0.90 for this model, but any errors or inaccuracies from this step will propagate through the pipeline and impact the final segmentation performance.

3. Novelty is limited. The only unique contribution is the lesion-aware masking technique, which, as mentioned earlier, is methodologically uncertain in how lesion regions are defined without relying on labelled data.

Minor:

1. The lesion-aware masking strategy is presented as the main innovation of the work. While it may encourage the model to learn meaningful features around pathological regions, it could also hinder its ability to capture the global context. A direct comparison of lesion-aware masking against a standard random masking baseline is essential in future work.

---

### Official Review · Reviewer_jy9q · 2025-10-27
**Self-Supervised Masked Autoencoders for Prostate Cancer Segmentation via Learned Representations**

**Rating:** 7
**Confidence:** 4

**Review:**

Summary
The paper presents a method for prostate cancer segmentation based on masked autoencoders (MAE). The proposed approach involves pretraining a visual backbone using tumor-aware masking, followed by fine-tuning a lightweight decoder for the segmentation task. Experimental results show that this strategy achieves significantly better performance than several state-of-the-art segmentation architectures.

Strengths:
- Prostate cancer segmentation is a highly relevant and clinically meaningful problem for the medical AI community.
- The paper is clearly written, well structured, and easy to follow.
- The proposed method demonstrates substantial improvements over multiple strong baselines.

Weaknesses
- It would strengthen the paper to include a comparison with a model using the same backbone (likely a ViT, as used in MAE) trained without pretraining. This would help to quantify the contribution of the MAE-based pretraining step.

Overall Assessment
The paper presents a solid and relevant contribution to prostate cancer segmentation. The proposed MAE-based pretraining strategy is effective, and the results show clear and meaningful improvements.

---

### Official Review · Reviewer_DyZi · 2025-10-31
**Review comments**

**Rating:** 4
**Confidence:** 5

**Review:**

The paper primarily combines two established concepts Masked Autoencoders (MAE) for self-supervised pretraining and a fine-tuning, with limited technical contribution. The proposed "lesion-aware masking strategy" is a straightforward mechanism (biasing mask retention based on label location) that does not constitute a substantial technical contribution.

The claims of novelty describe engineering choices or empirical strategies rather than a fundamentally new model architecture or learning algorithm. The core MAE architecture remains unchanged. My final rating goes to reject.

---

### Decision · Program_Chairs · 2025-10-31

**Decision:**

Reject

**Comment:**

The reviewers agree that the paper addresses a clinically relevant problem and is clearly written, but they differ in their evaluation of the contribution. While one reviewer finds the MAE-based pretraining effective and the results strong, others consider the methodological novelty limited and the self-supervised framing insufficiently justified.